# Optimized 3D Bioprinting Technology Based on Machine Learning: A Review of Recent Trends and Advances

**DOI:** 10.3390/mi13030363

**Published:** 2022-02-25

**Authors:** Jaemyung Shin, Yoonjung Lee, Zhangkang Li, Jinguang Hu, Simon S. Park, Keekyoung Kim

**Affiliations:** 1Biomedical Engineering Graduate Program, Schulich School of Engineering, University of Calgary, Calgary, AB T2N 1N4, Canada; jaemyung.shin@ucalgary.ca (J.S.); zhangkang.li@ucalgary.ca (Z.L.); jinguang.hu@ucalgary.ca (J.H.); 2Department of Mechanical and Manufacturing Engineering, Schulich School of Engineering, University of Calgary, Calgary, AB T2N 1N4, Canada; yoonjung.lee@ucalgary.ca (Y.L.); simon.park@ucalgary.ca (S.S.P.); 3Department of Chemical and Petroleum Engineering, Schulich School of Engineering, University of Calgary, Calgary, AB T2N 1N4, Canada

**Keywords:** 3D bioprinting, regenerative medicine, optimization, machine learning

## Abstract

The need for organ transplants has risen, but the number of available organ donations for transplants has stagnated worldwide. Regenerative medicine has been developed to make natural organs or tissue-like structures with biocompatible materials and solve the donor shortage problem. Using biomaterials and embedded cells, a bioprinter enables the fabrication of complex and functional three-dimensional (3D) structures of the organs or tissues for regenerative medicine. Moreover, conventional surgical 3D models are made of rigid plastic or rubbers, preventing surgeons from interacting with real organ or tissue-like models. Thus, finding suitable biomaterials and printing methods will accelerate the printing of sophisticated organ structures and the development of realistic models to refine surgical techniques and tools before the surgery. In addition, printing parameters (e.g., printing speed, dispensing pressure, and nozzle diameter) considered in the bioprinting process should be optimized. Therefore, machine learning (ML) technology can be a powerful tool to optimize the numerous bioprinting parameters. Overall, this review paper is focused on various ideas on the ML applications of 3D printing and bioprinting to optimize parameters and procedures.

## 1. Introduction

Tissue engineering has evolved from biomaterials science by integrating scaffolds, cells, and biomolecules to fabricate functional tissues. The main objective of tissue engineering is to develop three-dimensional (3D) artificial tissues and organs which can be used to augment, repair, and replace damaged or diseased tissue. Regenerative medicine is a broad field that includes tissue engineering and also involves research on the self-healing of our body by using its own or foreign materials to regenerate cells, tissues, or organs [1]. However, the terms “tissue engineering” and “regenerative medicine” have been used interchangeably, as the research area focuses on restoring impaired functions resulting from any cause, including congenital disabilities, disease, trauma, and aging.

On the other hand, personalized medicine is a novel field of treating individuals with optimized treatment [2]. Since everyone has different family medical histories and disease specificity, treating them with a homogeneous medication is not recommended [3]. People opt for personalized medicine over time, which has a low rate of immune rejection. Personalized medicine is different from precision medicine, commonly known as patient-specific therapy [4]. Precision medicine and patient-specific therapy are the creation of treatments that can be applied to groups of individuals with specific characteristics. Personalized therapy refers to patient-oriented treatment based on the clinical features of each patient by comprehensively considering the diversity, severity, and genetic characteristics of the symptoms.

Driven by tissue engineering, 3D bioprinting, an additive manufacturing process, is a pioneering technology that prints 3D structures with biocompatible materials including living cells (i.e., bioinks) [5]. While general 3D printing extrudes molten plastic that hardens to become a 3D object, 3D bioprinting is designed to print biological materials or bioink. A bioink is a combination of biomaterial, living cells, and hydrogel, and is the most widely used due to a highly hydrated environment for cell proliferation. Most matrix bioinks are naturally derived hydrogels, such as collagen, hyaluronic acid, or alginate. Moreover, materials widely used in bioink are gelatin methacrylate, collagen, poly(ethylene glycol), pluronic, alginate, and decellularized extracellular matrix-based materials. With 3D bioprinting, complex structures that cannot be fabricated using traditional manufacturing methods can be made and can alleviate the organ shortage problem in transplantation [6]. The bioprinted scaffolds can be turned into functional tissues or organs by encouraging cellular activities with various growth factors for regeneration. In addition, the bioprinted tissues or organs could be vital for surgeons to simulate complicated surgery. Three-dimensional bioprinting technology has the potential to replicate a variety of complex human tissues, leading surgeons and medical device companies to turn to it for their surgical training and testing needs. With more realistic surgical simulation models printed by the bioprinter, surgeons can improve their surgical techniques and reduce the chance of making mistakes in the surgery process.

There are various types of 3D bioprinting technologies. Extrusion-based bioprinting is the most widely used. It can produce scaffolds that do not contain organic solvents harmful to the human body and fabricate them precisely by applying pressure to the biomaterials [7]. Inkjet bioprinting has the advantage of printing various cells in a particular location precisely [8]. Stereolithography bioprinting uses a liquid photocurable resin that can be crosslinked with ultraviolet light [9]. The bioprinting process is broadly divided into three steps: pre-bioprinting, bioprinting, and post-bioprinting [10]. In the pre-printing stage, computer-aided design (CAD) is utilized with different types of software. Once the design is created in CAD, it will move to the slicing step for transitioning to the standard tessellation language (STL) file by generating G-code. G-code is a computer language that can tell a command to the printer. Appropriate design is related to how many tissues or organs can get the nutrients and oxygen. The initial design would provide valuable data to compare and analyze with post-printing results. Furthermore, the selection of proper biomaterials and bioprinters is important since these affect to rheological, biological, and degradation properties in the outcomes. Biocompatibility is an important factor when selecting the biomaterial, as it works with living tissues in the body or body fluids and the production of a toxic or immunological response can be lethal. In the actual printing phase, the defect detection of printed patterns in real-time would be investigated. In the post-printing step, suitable storage and continuous quality monitoring methods are investigated to maintain the best environment for cell growth. In this way, there are many challenges in 3D bioprinting and 3D printing when finding a suitable printing setup.

As a solution, machine learning (ML) techniques first appeared in the 3D printing field [11]. ML is a sub-branch of artificial intelligence (AI), and it is a technology that aims to realize functions such as the human learning ability in computers. ML algorithms can learn based on empirical data, make predictions, and improve their performance by themselves. These algorithms do not execute static program instructions but rather build an ideal model for making predictions or decisions based on input data. These computing technologies can be combined with cutting-edge medical technologies to create synergy. The ML-based model can achieve better performance compared to the physics-based model [12]. The traditional physics-based model has a challenge of non-linearities, varying parameters, and uncertainties. However, ML is highly flexible, has adaptable methods, and is excellent in prediction. A key question is how to choose between a physics-based model and a data-driven ML model. The answer depends on the problems to be solved. If there is no direct knowledge about the behaviour of a system, then mathematical models cannot be formulated and are unable to make an accurate prediction. However, if there are many example outcomes, the ML-based model can be used with training, validation, and testing datasets. In ML, the cost function can help analyze how well an ML model performs by comparing the predicted values with the actual values. The cost function is defined on an entire data instance, while the loss function is defined on a single data instance. The optimization algorithms in the cost function are used to find the optimal values for the model parameters by finding the minima of cost functions. As a nature of optimization, many printing parameters can be optimized to achieve the best design and printing environment relative to a set of prioritized criteria or constraints.

The ML, applied for additive manufacturing or standard 3D printing, has already gained much attention due to its ability to fabricate parts with optimized printing parameters [13]. The ML technology was effectively used in the overall 3D printing processes to improve numerous parameters (e.g., filament size, melting/bed temperature, printing speed, layer thickness, infill density, and nozzle size) [14]. However, compared to standard 3D printing, few studies on using ML in 3D bioprinting processes have been found, despite many potentials. While new technologies such as telemedicine, healthcare metaverse, and medical wearables are connected organically, 3D bioprinting has emerged as an innovative medical application in biomedical engineering. More finely printed outcomes due to ML can help test drugs more safely, decrease the amount of animal testing, and even replace or repair damaged organs with high printing fidelity in the future. Additionally, increasing attention on bioprinting research based on ML will accelerate advancements in regenerative medicine. This review paper provides an overview of the use of 3D bioprinting systems in the time frame of 2015 to 2021 and recent studies applying ML to 3D bioprinting in the time frame of 2016 to 2021. Based on this, researchers can inspire new ideas on how ML can be used to improve 3D bioprinting.

## 2. Machine-Learning Principles Used in 3D Bioprinting

The 3D printing technology significantly impacts various industries and research fields (e.g., manufacturing, medical, robotics, automotive, aerospace, and education) [15]. In tissue engineering, 3D bioprinters can print human tissues and organs with bioinks consisting of biomaterials and cells [16]. They can create complete substitutes for damaged tissues and fabricate microscale tissues or organoids for disease study and drug discovery [17,18].

### 2.1. Materials and Modalities of 3D Bioprinting

Regardless of specific 3D bioprinters, bioinks in the bioprinting process are indispensable components [19]. Bioinks consist of cells and hydrogel pre-polymers for the formation of tissue scaffolds. The cellular component should be selected as the essential consideration. According to the methods and applications of printing, the selected cells for bioink should mimic the physiological state in vivo [20]. Then, the cells can maintain their functions in biomaterials after they are printed, as shown in Figure 1.

The cell source includes primary cell lines from human and animal tissues, stem cells, etc. [22]. Primary cells are directly taken from human or animal tissue using enzymatic or mechanical methods [23]. Stem cells are the proper cell sources for bioprinting since they increase the proliferative capacity and sufficiently mimic the function of the original cells. Even though most of these cells can be purchased easily, specific cells are challenging to cultivate due to the limitations of the current bioprinting systems. Thus, the labs that mainly focus on bioprinting develop advanced cells for commercial use or research.

Biomaterials: The viscosity and rheological behaviour of biomaterials influence the quality and shape of printed constructs. Therefore, the selection of appropriate biomaterials before printing is essential. Hydrogels have become a popular material to pattern cells in 3D bioprinting due to their high biocompatibility and processability. Hydrogels are necessary material, especially in bioprinting [24]. The terminology hydrogels is composed of “hydro” (water) and “gel” and refers to aqueous (water-containing) gels [25]. They are polymer networks that are insoluble in water and swell to an equilibrium volume but retain their shape. Hence, hydrogels directly contact cells to provide a growing environment like an extracellular matrix and dominate the mechanical and biocompatible properties of the printed object. The extracellular matrix forms an integral part of the scaffold to induce cell proliferation, thereby leading to new tissue formation. Printable and crosslinkable properties should normally characterize a hydrogel pre-polymer solution for bioprinting. This means the pre-polymers are usually required to have shear thinning properties or the carbon–carbon double bond structures, as shown in Figure 2. In addition to the printability, the pre-polymer solution should be a material compatible with living tissues as a biocompatible property. In terms of crosslinking, the strategies to transfer pre-polymer solution to hydrogels include physical crosslinking, chemical crosslinking, and enzymatic crosslinking. Physical crosslinking uses interactions other than a covalent bond, such as coordination bonding, hydrogen bonding, ionic interaction, or van der Waals interaction. Thermoplastic elastomers are the representative material that undergo physical crosslinking, and thermosetting polymers are made from chemical crosslinking. Chemical crosslinking refers to the intermolecular or intramolecular combining of two or more molecules by a covalent bond. The mechanism of different crosslinking methods comprises the non-covalent/covalent interactions to combine polymer chains and make them form a stronger network. According to the crosslinking techniques, different bioprinters are recommended, as shown in Figure 2.

Pre-polymers that form hydrogels in bioprinting can be classified into biopolymers and synthetic polymers. Biopolymers are natural polymers produced by the cells of living organisms. They are derived from humans, animals, and plants with the examples of cellulose, alginate, collagen, and chitosan [26]. Usually, these natural biopolymers are suitable for extrusion printing due to non-covalent interactions. This results in excellent rheological properties such as shearing shining and thixotropy. The advantage of the biopolymers is they can naturally cover the surface of eukaryotic cells and combine with proteins. These characteristics can form a natural extracellular matrix, resulting in them having high biocompatibility and cell affinity.

On the other hand, synthetic materials are derived from petroleum oil and are made by scientists and engineers [27]. The chemical and physical properties as crosslinking rates and mechanical properties can be precisely designed. Polyethene glycol and polyacrylamide, polyacrylamide, and poly (N-isopropyl acrylamide) are commonly used in 3D printing. The ink containing polymer or monomers for hydrogel printing should have suitable rheological or light-curing properties [21]. The printable hydrogels must have good biocompatibility and provide an appropriate environment for the survival and growth of cells. In addition, hydrogels should have excellent mechanical properties. Some biomaterials need to be mixed with cellulose, polyvinyl alcohol, cellulose fibrils, and chitosan to improve the viscosity and shear-shining properties via molecular interactions. There is another method to refine the light-curing property of biomaterials [28]. Gelatin methacrylate and hyaluronic acid methacrylate modify biopolymers by grafting carbon–carbon double bonds on the polymer chains. Unlike natural biopolymers, these modified polymers are more suitable photopolymerization print modalities due to the existence of carbon–carbon double bonds. These polymers simultaneously have the advantages of biopolymers and synthetic polymers and can be crosslinked under ultraviolet or visible light. However, these materials are not enough to fabricate sophisticated and complex human tissue/organs due to the rapid development of printing technology. Seeking a new hydrogel for 3D bioprinting remains an ongoing and exciting task.

Modalities: Four primary 3D bioprinters were categorized and explored in this section [8,29,30,31,32]. As shown in Figure 3, there are extrusion-based bioprinting, scaffold-free bioprinting, inkjet bioprinting, and laser-induced forward transfer (LIFT) bioprinting. Extrusion-based bioprinting is a relatively simple 3D bioprinting method, which uses shear stress to extrude bioinks to deposition 3D structures [33,34]. Bioinks for extrusion-based bioprinting usually consist of alginate or fibrin polymers. They are flexible enough after shearing smoothly extruded from the nozzle. Furthermore, they are integrated with cellular adhesion molecules, which support the physical attachment of cells and maintain structural stability rigidly after printing.

As shown in Figure 3A, the bioink-containing cells are put into the syringe. The bioinks are extruded through nozzles by shear force to create 3D structures. The extruded bioink forms the specific shape that was expected to be printed through a CAD program. Although there are many 3D printings processes, most of them rely on CAD [35]. CAD is the process of designing an object using computer software and is an essential component of 3D printing. Slicing a 3D model means taking the design in STL format and slicing it into individual layers. The software then generates the tool path (G-code). This process is important for translating the 3D drawing into a language that a 3D printer can understand and print. It commands the 3D printer how much material it needs to deposit and where the material should be deposited.

Scaffold-free bioprinting has gained extensive attention due to the advantages of high cell content and shorter restructuring time than the scaffold-based technique [30]. Scaffold-free approaches have been used in clinical practices to restore damaged tissues such as bone, cartilage, and ligaments. A notable benefit of the scaffold-free strategy is that cells naturally create a tight intercellular connection [36]. This method is used to print tissues or organs with a high concentration of cells. Scaffold-free bioprinters generate aggregates or rods entirely composed of cells. These aggregates form larger constructs as time goes on based on tissue liquidity and fusion (Figure 3B) [30,36]. The self-assembly of cells resulting in more giant constructions depends on cell and molecular interactions.

Inkjet printing is a relatively low-cost technology since commercial printers (i.e., office inkjet printers) have been transformed into inkjet bioprinters [4]. The principle of 3D inkjet bioprinting is like traditional inkjet printing, in which the ink is applied using the piezoelectricity or heating strategy. Three-dimensional inkjet bioprinting creates biological structures by gradually depositing bioinks in a specific place (Figure 3C) [30,31]. The investigation of the combination of individual ink drops is the most crucial part of 3D inkjet bioprinting. As the dimension of printable droplets is tiny, the bioprinting of a large structure such as an organ might be a challenge; however, the small droplet size is of great merit to the resolution of printed designs.

LIFT bioprinting is similar to inkjet bioprinting [37]. The difference is that a laser beam is used as a driving force to project bioinks from a donor thin film composed of biopolymer and toward the receiving platform in LIFT bioprinting (Figure 3D) [27,37]. LIFT bioprinter does not have nozzles; hence, the clogging issues in nozzles can be naturally solved. Even though the setup of the LIFT printer is more expensive and is not fully automated, this printer can print a more accurate structure at a faster rate. Overall, these four bioprinters have advantages and disadvantages. It is greatly encouraged to select the most suitable printing method according to the characteristics of the tissue/organs, the cell, and biomaterials.

### 2.2. Machine Learning

ML was coined in 1959 for the first time, and it is defined as the study of computer algorithms that can automatically improve through experience using data. ML is a subset of AI, and it can predict or decide without being explicitly programmed to do so. Types of ML can be defined as four broad categories, according to the nature of the signal or feedback available to the learning system: supervised ML, unsupervised ML, semi-supervised ML, and reinforcement ML. Each ML type will be explained in the following paragraph in detail. Shin et al. [38] used supervised ML to classify healthy and infected leaves in the field and developed their algorithms with a classification accuracy of 95.59%. Furthermore, they used deep learning (DL) algorithms (e.g., AlexNet, SqueezeNet, GoogLeNet, etc.) to detect and classify the healthy and infected leaves with better accuracy than ML. DL is a type of ML, and DL algorithms use more complex multi-layered neural networks. The information is transferred from one layer to another over connecting weighted channels. Generally, the training and testing dataset is split into specific ratios. For very large datasets, 80% for training and 20% for testing; however, for small datasets, 60:40% to 70:30% is required [39]. In terms of data points, 1000 examples are required at a minimum, and normally, 10,000 examples are needed for optimal results [40]. ML and DL have gained much attention due to their ability to predict and decide optimally by reducing the workload. This section will introduce the four types of ML and DL and explore the research in which ML has been applied to 3D printing and 3D bioprinting.

Supervised ML: Supervised ML is the most common ML process, and needs labelled datasets and creates desired outputs for linked inputs with mapping [41]. Supervised ML has been utilized for email spam filtering, face recognition in public areas, and the diagnosis of diseases with image processing. Supervised ML has a training dataset, and the algorithm predicts the expected labels based on the remaining validation dataset. Data can be collected from prior experience for supervised ML, or data output can be generated. The experience can optimize performance criteria, and it has the advantage of helping to solve various problems. Supervised ML can be categorized into classification and regression depending on whether the labels are discrete or continuous [42]. Classification predicts the outcomes based on categories or labels, but regression usually predicts the results based on the characteristics of the data. There are binary and multi-classes in classification, which has two or several classes to predict, respectively. There are several algorithms in supervised ML such as naïve bayes, decision tables, support vector machines, random forest, *k*-nearest neighbour, etc.

The biggest challenge is to obtain the required dataset; particularly, it requires a lot of computation time in the case of big data. In addition, the pre-processing of data is no less than a big challenge. Despite these issues, supervised ML can be placed under full control of the process. The review paper by Goh et al. [13] about ML in additive manufacturing organized the algorithms of supervised ML in 3D printing, and they are shown in Table 1 [43,44,45,46,47,48,49,50,51]. The supervised ML algorithms in additive manufacturing, which includes bioprinting, are investigated in Section 2.3.

Unsupervised ML: Unsupervised ML is used with ambiguous datasets that are not distinguished and structured in the category [52]. A larger amount of data can be effectively used in unsupervised ML. Since there are no training and labelled datasets, unsupervised learning algorithms should initially discover occurring patterns in the training dataset [53]. Unsupervised ML can be utilized in the pre-processing step to find representative features before using the supervised ML with robust features. Unsupervised ML creates ambiguous input dataset structures to learn more information related to the data [54]. Such unsupervised ML characteristics can be applied in various fields, including grouping clients with similar tendencies in the business system, classifying similar flower types or animals, and detecting abnormal access on the network. Types of unsupervised ML include clustering, anomaly detection, association, and autoencoder. Additionally, according to the review paper by Goh et al. about ML in additive manufacturing, *k*-means clustering, self-organizing maps, and restricted Boltzmann machines are used in 3D printing as unsupervised ML [13].

Semi-supervised ML: Semi-supervised ML is a combination of supervised and unsupervised ML [55]. While training, semi-supervised ML integrates a small amount of labelled data with a large amount of unlabelled data. In the classification, unlabelled data are added to the existing supervised learning data to improve performance. In clustering, labelled data can aid in deciding which cluster to put new data into. Semi-supervised ML is widely used to boost the performance of classifiers by utilizing unlabelled data when the number of labelled data is small. As the ability of ML grows, labelling has shifted from simple and repetitive tasks to complicated works that require a lot of professionality. Thus, semi-supervised ML has been highlighted in many ways by handling data that are relatively unlabelled.

For this semi-supervised ML to properly improve the performance, several assumptions are required [56]. The first presumption is the smoothness. This assumption is that if the inputs *x*_1_ and *x*_2_ in a region with a high probability density are close, the corresponding labels *y*_1_ and *y*_2_ must be relative [57]. The second assumption is low density, in which the decision boundary of the model does not pass where the probability density of the data is high. Finally, the various assumptions whereby the high-dimensional input data are placed along with a specific structure named a manifold in a low-dimensional space. In short, if data values close to the input space are collected and considered a cluster, this is a smoothness assumption [58]. Suppose data points in a high-probability-density area are regarded as clusters. In that case, it becomes a low-density assumption, and data points on a low-dimensional manifold are examined as a cluster and it becomes a manifold assumption. For these principles, making good use of the assumptions explained above, unlabelled data can be utilized meaningfully. Popular approaches include self-training, minimum-margin, and perturbation-based algorithms. Self-training, the most fundamental semi-supervised ML, uses a simple pseudo-labelling method [59,60]. Here, only labelled data are processed at the onset of training, and then unlabelled data are gradually pseudo-labelled and used for further training.

Reinforcement ML: Reinforcement ML is to learn by providing rewards or punishments based on the outcome of the actions [61]. Instead of training data used in supervised/unsupervised ML, it rewards the consequence of activities tailored to a given state, improving performance. Therefore, training processes are repeated to receive better rewards in reinforcement ML. It requires data generated from environments or simulations, clearly defined rules of reward and punishment, and sufficient processing power. In this regard, reinforcement ML includes supervised and unsupervised ML algorithms for linking inputs and outputs.

Reinforcement ML can be implemented through a Markov decision process that performs sequential actions in a known environment [62]. It is different from general supervised ML. It does not require knowledge of the Markov decision process; it does not explicitly correct wrong behaviour and it is characterized by focusing on online performance. In other words, reinforcement ML strikes a balance between exploring areas that have not yet been investigated and using the already known knowledge. It requires only the current state and action, and the reward or punishment is determined based on this action. This learning method is inspired by behavioral psychology and helps solve problems in various disciplines, including operational science, information theory, and simulation-based optimization.

Deep Learning: DL is a type of ML, and both algorithms learn automatically by themselves [38]. When it comes to ML in recent years, it often refers to ML technologies other than DL. It is necessary to look at the process of ML to understand DL. ML goes through the process of collecting data, processing variables, training models, and evaluating them. Here, DL is to achieve good performance even without the variable processing step. That does not mean that DL consistently outperforms ML. It must be appropriately chosen according to the situation, the problem, and the data size. DL is suitable when there is a large data size; simultaneously, it requests a high-end computer and considerable learning time. Data augmentation is helpful to solve the problem of large amounts of data [63].

Moreover, Guan et al. [64] provided the bioprinter with optimal parameters for DL algorithms developed to compensate for cell-induced light-scattering effects. Their study reduced the trial and error to optimize parameters for each specific printable structure due to the light scattering of cells in the bioink, a chronic problem of digital light-processing-based bioprinting methods. DL techniques are effective when there is no domain knowledge about the data, and the understanding of the variables is insufficient. In addition, DL has outstanding capabilities in complex problems such as image classification, natural language processing, and speech recognition.

### 2.3. Bioprinting Process with Machine Learning

A successful bioprinted tissue and organ usually undergo three steps: pre-printing, printing, and post-printing [64,65]. Figure 4A describes an at-a-glance flow of printing an organ. Bioimaging and CAD software were used in the pre-printing process to design a blueprint from generating an STL file to slicing an STL file into binary slices. In the printing stage, adding bioinks, patterning the slice of one layer, and repeating the process until the whole pattern is finished are the processes to create a tissue/organ. The schematic of designing the printable bioink is shown in Figure 4B [66]. In the post-printing process, the extra inks should be removed and washed with PBS. The cell in bioinks would grow and proliferate to mimic the functions of the organ in an incubator. However, creating a fully functional and large-scale organ remains a challenge using 3D bioprinting even though many advancements have been made. Accordingly, ML technology could be merged with these three steps to improve the fidelity of complex printing.

Pre-printing: The pre-bioprinting phase is crucial since it relates to the properties of the bioprinting structure. Planning and determining the appropriate conditions helps achieve anatomically perfect tissue models and ensures the high quality of the cells [64]. Recently, 3D bioprinting research using ML technologies has been increasing. This section explores ML algorithms applied to the pre-printing process in bioprinting.

Determining the appropriate bioink for bioprinting, which relates to printability and biocompatibility, is crucial. Lee et al. [66] developed a printable bioink based on an inductive logic programming (ILP) methodology and multiple regression. Multiple regression was used to predict the printability of bioink. They prepared several bioink compositions with three types of native collagen and atelocollagen, respectively, with hyaluronic acid and fibrin. They evaluated the frequency-dependent storage (G′) and shear stress of bioink and printability, including fidelity of the shape and nozzle clogging. As a result, the higher shape fidelity needs higher G′, and extrusion needs low shear stress. This research conducted by Lee et al. [66] is expected to be used as the guideline for the development of bioink. Shi et al. [67] utilized ML to optimize the design parameters using multi-objective optimization problems for piezoelectric drop-on-demand. This printing method has the challenges of low precision and stability for printing, large droplet size, and slow printing speed. Fully connected neural networks were used for the connectivity in printing parameters. Moreover, they explored the hybrid multi-sub gradient descent bundle method with an adaptive learning rate algorithm. As a result, they printed a single droplet reliably and increased the print speed from 0.88 to 2.08 m·s^−1^ by improving print precision and stability. This finding enables the realization of precise cell arrangements and complex biological functions and guides the establishment of bioprinting platforms.

Actual printing process: ML algorithms in the printing process can contribute numerous essential parameters while printing [13]. For example, process and part quality optimization and prediction of various anomalies can be solved. In actual printing, it is important to select a bioprinter that is suitable for research and the environment, since the characteristics of bioinks are different depending on the bioprinters.

There are four primary bioprinting modalities: extrusion-based bioprinting, scaffold-free bioprinting, inkjet bioprinting, and LIFT bioprinting. The details of these modalities have been described in Section 2.1. In all 3D bioprinting modalities, ML can suggest a more effective way of selecting bioinks, 3D bioprinters, and software. Complex instructions can be learned and trained by the ML approaches. Considering that, ML will improve the fabrication process via optimizing existing parameters and predicting possible conditions. Jin et al. [68] developed an anomaly detection system to classify imperfections for hydrogel-based bioink based on convolutional neural networks. Images were processed as small image patches for a grid, gyroid, rectilinear, and honeycomb shape. This research envisions high-quality tissue composition through a real-time autonomous correction in the 3D bioprinting process.

For extrusion bioprinting, the parameters include the printed filament diameter (d), nozzle diameter (D), nozzle length (L), the apparent viscosity of bioinks (m), stage moving speed (v), and the shear stress (S) [68,69]. Modelling the relationships can facilitate a more suitable parameter selection to refine the printing fidelity and reduce bioprinting time. There have been several reports investigating the relationship among these parameters. For instance, the printed filament diameter should be determined by considering the nozzle diameter (D), the apparent viscosity of bioinks (m), nozzle length (L), and stage moving speed (v). In addition to the printed filament diameter that affects the resolution, embedding appropriate cells in the scaffold is indispensable and critical for bioprinting. Nonetheless, the studies exploring the parameters related to cell viability and proliferation were still insufficient based on ML and DL. As shown in Figure 5, the printing parameters are fed into the neural network as data for training. Then, corresponding results (e.g., cell viability and cell proliferation) are used tune the parameters in the algorithms. The neural network architecture can predict the printed outcomes based on the unseen data.

For LIFT bioprinting, the laser time and intensity negatively affect cell viability; however, these parameters are crucial to the biomaterial’s crosslinking and mechanical properties in the case of hydrogels. After training the neural network on parameter (laser)-outcome (cell and mechanical properties) data, the parameters would be optimized for balancing the degree of crosslinking and cell viability. Based on the trend mentioned above, ML could be a useful tool in developing 3D bioprinting in the future.

The convolutional neural network can identify defects in diverse manufacturing domains [70,71,72]. In the bioprinting process, ML can monitor the printing process in real-time and improve printing fidelity. This can detect approximate dimensions and curved layers and correct the wrongly positioned bioinks and microstructure errors. The convolutional neural network has a significant advantage on position-invariant features when extracting the representative features regarding the edges and lines on the objects [54]. Jiang et al. [54] described an example of using a convolutional neural network to detect the defects and errors in the 3D bioprinting process (Figure 6). High-quality images were fed to the convolutional layers. Then, the data were analyzed and labelled with different classifications by the model. Lastly, the classification algorithms detected the imperfection of bioprinting.

Post-printing: Post-printing is the most crucial step in determining the quality of printed outcomes among bioprinting procedures [73]. Common issues in post-printing are inconsistency between the expected printed outcome and the actual results in terms of design and cell proliferation [74]. Even if the blueprint is appropriately designed and fabricated before post-printing, if the printed products are not adequately managed, there are many limitations to being delivered to the actual patient or surgeon and being utilized. To make the printed 3D tissue/organs functional, they should be overseen under proper tissue maturation and could take as long as eight weeks [75]. Storing under a controlled environment (i.e., pH, temperature, gas concentration, nutrient supply, waste removal, and physical stimulation supply) helps maintain cell growth and viability [76]. Experimental verification of potential maturity factors is essential for promoting tissue maturation and maintaining tissue properties. Therefore, post-printing should be processed by integrating possible knowledge of the cells, bioprinters, and printing properties. In a situation where these conditions are harmonically combined, ML is a suitable state-of-the-art technique.

In biofabrication, scaffolds play a role in creating a microenvironment in which cells can grow and differentiate in tissue structures by combining cells, growth factors, and biomaterials. Therefore, scaffolds are generally complex since they involve various roles, such as an active interaction between cells and biomaterials, cell adhesion, and transport of nutrients for cell growth. ML can predict the accuracy of success rate for building scaffolds whereby good-quality outcomes can be achieved [77]. Furthermore, ML quantitatively evaluates the printability of attaining a scaffold specialized for the individual’s skeletal structure and optimizes the advantages for various printers [15]. Ruberu et al. [15] used a probabilistic model to suggest a better printer setup with a scoring system for filament. The training data were printing parameters (e.g., extrusion pressure, bioink temperature, and printing speed) and different bioink concentrations. They evaluated and optimized the printability of gelatin methacrylate and hyaluronic acid methacrylate bioinks using Bayesian optimization. To quantitatively evaluate the printability, a pore structure on layer stacking of filament morphology and layer stacking was created with a scoring system during extrusion.

## 3. Challenges and Future Prospects

ML is a branch of AI, a computer science methodology that aims to identify complex patterns in data that can be used to make predictions or classifications on unseen data. In 3D bioprinting, ML can indicate the optimized parameters when printing the most suitable tissue/organs for a specific patient by improving the printed outcomes’ quality and detecting the anomalies. This progress dramatically saves printing time and bioinks, which can be wasted in trying numerous trials and errors. Moreover, it is beneficial for researchers to prepare the medium in which cells grow optimally and simulate the structure of human tissues and organs.

Nonetheless, few studies are reported in bioprinting based on ML compared to general printing with ML. Perhaps because of its biomedical nature, it is challenging to collect massive amounts of data. ML requires a large amount of data when training and testing to achieve high classification accuracy. However, as open source expands, a variety of data will become publicly accessible. The prospect is highly bright, as it is expected that data collection will become much easier as researchers share each other’s data in the cloud. For example, patient-specific data from all individual 3D models are generated based on computed tomography and magnetic resonance imaging data and then digitally using computer-aided manufacturing software. To achieve optimal treatment results, the patients should not accept a unified, standardized treatment; however, personalized medicine should be obtained.

Another question is how to deal with transferability for AI. ML models that are trained are primarily dependent on mathematical features of the training data, so it is not guaranteed to operate consistently with equivalent abilities. To increase AI transferability, therefore, data selection should be optimized through data scaling, including normalization or standardization. Additionally, transfer learning is effective in that AI quickly adapts to similar domains. Likewise, many critical questions are centred around transferability between different models and data quality. Advanced ML, the combination of traditional physics models, and digital twin are required to solve these challenges to develop robust 3D bioprinting processes. The digital twin is the virtual model, which embodies the experimentation identical to the real world. Experiments with computer simulations in a virtual environment are named in silico experiments. An et al. [78] described the future of bioprinting associated with ML. Specifically, they focused on finding the missing links between big data and a digital twin, leading to bioprinting’s digital and in silico experiments. Collecting publicly available bioprinting-related big data or significant data curation is the first step in solving the complexity of bioprinting and leading to in silico experiments. Establishing the digital twins of tissues/organs requires a good understanding of biological functionality and replication for cellular resolution and cell properties. The fidelity of the bioink can make the future 3D bioprinting process be in silico or digital. Finally, big data and digital twins would increase efficiency by enabling bioprinting simulations, making the bioprinting industry digitalized and automated. AI has been employed by various industries, and ML is proving its ability to adapt to new opportunities. The continuing development and optimization of technologies will help shape new trends in 3D bioprinting.

## 4. Conclusions

Three-dimensional bioprinting has made remarkable progress in proving the feasibility of organ transplants and improving patients’ treatment methods [5]. Personal genetic information varies; hence, understanding personalized medicine is essential. Personalized medicine is a promising and emerging medical practice that actively utilizes an individual’s genetic profile to guide disease prevention, diagnosis, and treatment decisions [4]. Here, 3D bioprinting can create complete replacements of the tissue/organs for transplant surgery. Furthermore, 3D bioprinted organs/tissues using individual decellularized extracellular matrix and stem cells could be used for patient-specific transplantation. However, fabricating fully functional tissues/organs still has many challenges due to limitations such as incarnating vasculatures for the solid organs (e.g., liver, pancreas, and spleen and adrenal glands). Moreover, novel techniques are demanded to process and understand these complex data collected from personalized medicine diagnostic approaches [79]. Fortunately, computer science has developed swiftly to progress techniques that enable the storage, processing, and analysis of these complicated datasets, a feat that traditional statistics and early computing technologies could not accomplish. In this regard, this review paper explored the contents of 3D bioprinting and ML, respectively, and examined the recent studies that ML has been applied to for the development of 3D bioprinting. ML can optimize the bioprinting process and parameters to increase printing fidelity. Hence, combining 3D bioprinting with ML techniques would be a great research area to accelerate future development.

## Figures and Tables

**Figure 1 micromachines-13-00363-f001:**
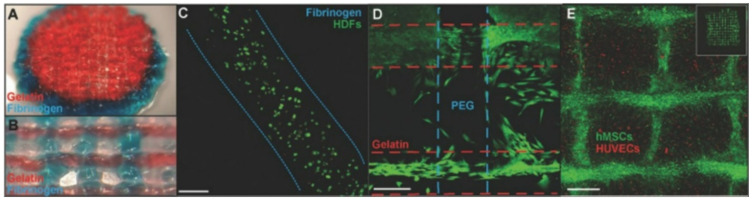
Cell-laden bioprinting. (**A**) Polyethylene glycol ending in two reactive groups (PEGX)-gelatin (red) and PEGX-fibrinogen (blue) coprinted cylinder, 15 mm diameter. (**B**) Coprinting with inner structure pattern ≈650 μm diameter. (**C**) Stained with Live/Dead assay at one day to check cell viability. (**D**) Coprinted structure with human dermal fibroblasts cell-laden printing. (**E**) human umbilical vein endothelial cell (HUVEC)-laden printing and human mesenchymal stem cells (hMSC) (cell tracker green) spread into open spaces of construct and onto printed bioink strands at day 4 (adapted from [21] with permission).

**Figure 2 micromachines-13-00363-f002:**
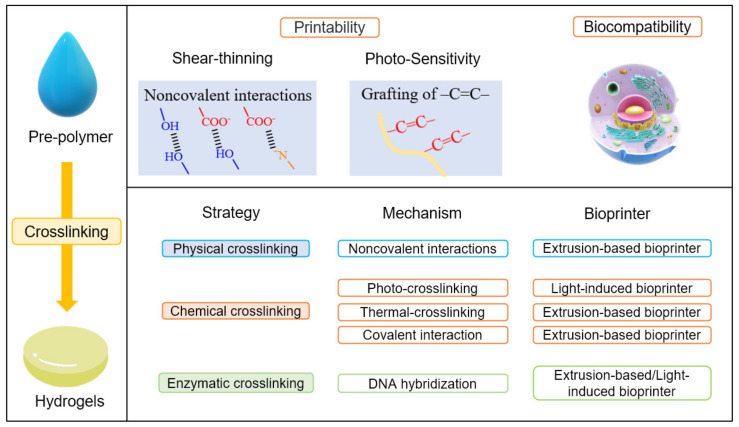
The crosslinking methods in 3D bioprinting of polymeric hydrogels.

**Figure 3 micromachines-13-00363-f003:**
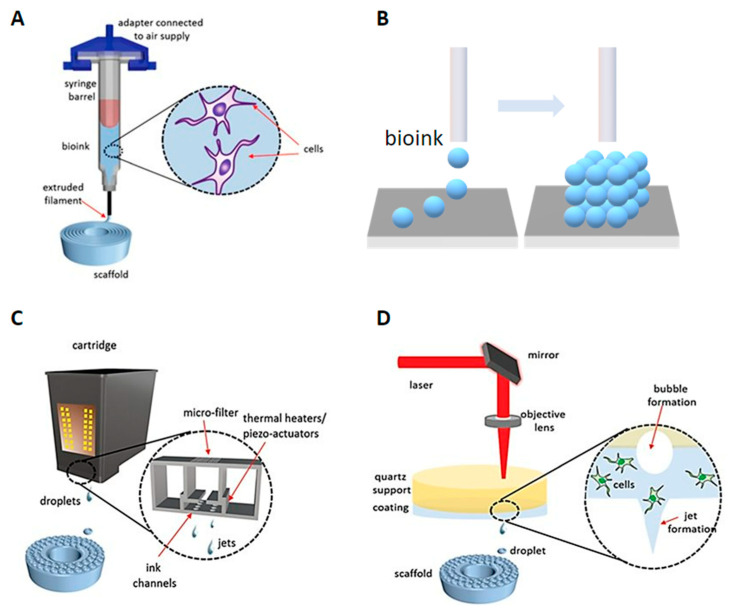
Modalities for bioprinting technologies. (**A**) Extrusion-based bioprinting. (**B**) Scaffold-free bioprinting. (**C**) Inkjet bioprinting. (**D**) Laser-induced forward transfer bioprinting (adapted from [32] with permission).

**Figure 4 micromachines-13-00363-f004:**
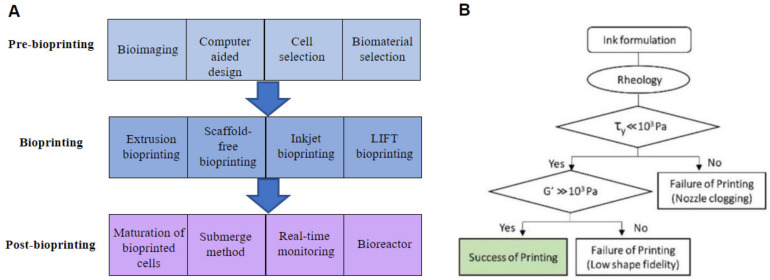
(**A**) Three stages: pre-bioprinting, bioprinting, and post-bioprinting and considerations in fabricating the tissue/organs constructs. (**B**) The schematic of developing the bioink (adapted from [66] with permission).

**Figure 5 micromachines-13-00363-f005:**
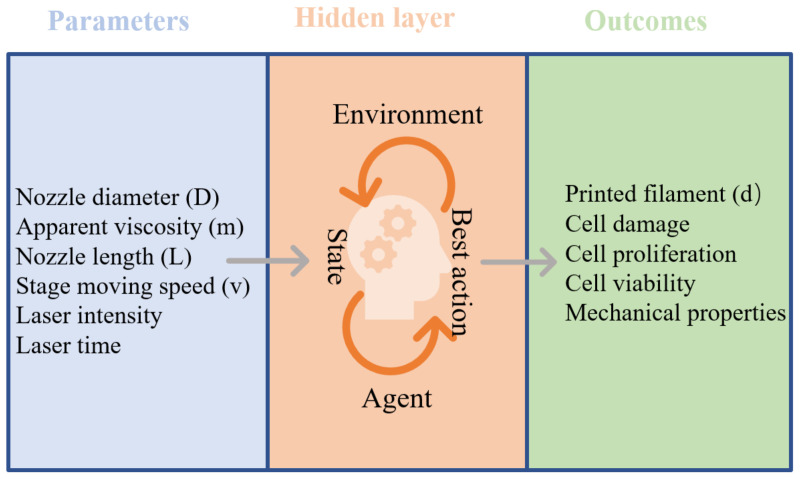
Examples of consideration in bioprinting for a neural network optimization.

**Figure 6 micromachines-13-00363-f006:**
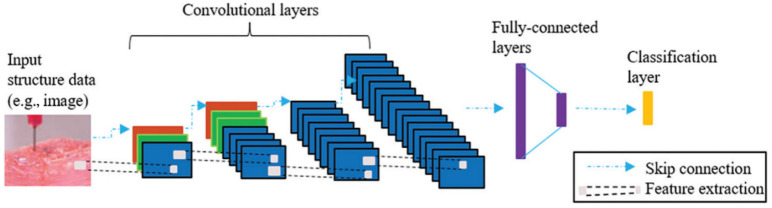
A framework of convolutional neural networks used in bioprinting (adapted from [54] with permission).

**Table 1 micromachines-13-00363-t001:** Supervised machine-learning algorithms used in general 3D printing.

Algorithm	References
Naive bayes	[43]
Decision tree	[44]
Convolutional neural network	[45]
Genetic programming	[46]
Long short-term memory	[47]
Particle swarm algorithm	[48]
K-nearest neighbour	[43]
Radial basis function	[49]
Siamese neural network	[50]
Support vector machine	[51]

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
