# Peer review of "Optimized 3D Bioprinting Technology Based on Machine Learning: A Review of Recent Trends and Advances"

_micromachines, 2022, doi:10.3390/mi13030363_

Round 1

Reviewer 1 Report

This is an interesting topic for using ML in 3D bioprinting. Here are some comments to improve the paper.

  • In the introduction there is a paragraph that is not related to this paper and belongs to the description of the journal format and should be removed
  • In the introduction authors should briefly define the 3D printing, 3D bioprinting, bioink also it is good to mention different types of the bioinks. Here is a reference that can be useful

https://doi.org/10.1016/j.mtbio.2019.100008

also it would be good  to explain (define) briefly the concept of ML in the introduction

  • Authors should mention the novelty of this review compared to the previous ones such as https://dx.doi.org/10.18063%2Fijb.v6i1.253

  • All abbreviations should be double checked
  • Authors should provide more examples of using machine learning for 3D bioprinintg

Here are a few more examples

https://doi.org/10.1016/j.jmrt.2021.07.004

  • for the work by Jin et al. authors should mention more details about this work: https://doi.org/10.1021/acsbiomaterials.0c01761

the same for the work by Rubero et al.

https://doi.org/10.1016/j.apmt.2020.100914

  • it would be better if authors changed the section 3 title to “challenges and future prospects” and also add another section at the end entitled “ conclusions” to state the summary and conclusions of this review
  • Permissions for the figure and related copyrights should be mentioned in the captions of the figures that are adapted from other published works

Reviewer 2 Report

The following review paper reports recent trends and advances in the field of optimized 3D bioprinting technology based on machine learning. It addresses the need for organ transplants since the number of available organ donations for transplants is nowadays stagnated worldwide. In this context, regenerative medicine has been developed to make natural organ or tissue-like structures with biocompatible materials.

The review paper is interesting and reports various ideas on machine learning applications of 3D printing and bioprinting to optimize the parameters and processes.

The language is satisfactory. My only remark is about the need to insert in lines 106 and 107 the time frame considered by the authors for the overview of the use of 3D bioprinting systems but also for  the studies involving machine learning to 3D bioprinting.

Round 2

Reviewer 1 Report

Authors addressed my comments.

Author Response

Thank you for accepting the manuscript